# Update on Molecular Diagnostics in Thyroid Pathology: A Review

**DOI:** 10.3390/genes14071314

**Published:** 2023-06-22

**Authors:** Bayan Alzumaili, Peter M. Sadow

**Affiliations:** Departments of Pathology, Massachusetts General Hospital and Harvard Medical School, Boston, MA 02114, USA; balzumaili@mgh.harvard.edu

**Keywords:** thyroid, molecular diagnostics, BRAF, RAS, thyroid cancer, thyroid carcinoma

## Abstract

Thyroid nodules are quite common, and the determination of a nodule of concern is complex, involving serum testing, radiology and, in some cases, pathological evaluation. For those nodules that raise clinical concern of neoplasia, fine needle aspiration biopsy is the gold standard for evaluation; however, in up to 30% of cases, results are indeterminate for malignancy, and further testing is needed. Advances in molecular testing have shown it to be of benefit for both diagnostic and prognostic purposes, and its use has become an integral part of thyroid cancer management in the United States and in several global nations. After The Cancer Genome Atlas (TCGA) consortium published its molecular landscape of papillary thyroid carcinoma (PTC) and reduced the “black matter” in PTC from 25% to 3.5%, further work ensued to clarify the remaining fraction not neatly attributed to the *BRAF*^V600E^-like or *RAS*-like phenotypes of the TCGA. Over the past decade, commercial molecular platforms have been refined as data accrues, and they increasingly cover most genetic variants of thyroid carcinomas. Molecular reporting focuses on the nodule tested, including related clinical information for that nodule (size of nodule, Bethesda category, etc.). This results in a comprehensive report to physicians that may also include patient-directed, clear language that facilitates conversations about nodule management. In cases of advanced or recurrent disease, molecular testing may become essential for devising an individual therapeutic plan. In this review, we focus on the evolution of integrated molecular testing in thyroid nodules, and how our understanding of tumor genetics, combined with histopathology, is driving the next generation of rational patient management, particularly in the context of emerging small, targetable therapeutics.

## 1. Introduction

Increased knowledge about the molecular underpinnings of thyroid neoplasia has transformed the approach to diagnosis, prognosis, and therapeutics. Molecular profiles have become an integral part of thyroid cancer management. These sensitive and specific ancillary tools have been useful for subclassifying indeterminate (for malignancy) thyroid nodules by cytology (Bethesda III or IV) or ambiguous nodules by light microscopy (tumors of uncertain malignant potential), resulting in more optimal patient care [1]. The current (fifth edition) World Health Organization criteria categorizes thyroid tumors into follicular-derived neoplasms (benign, low-risk and malignant), C cell-derived carcinomas (medullary thyroid carcinoma; MTC), mixed medullary and follicular cell-derived carcinomas, and other rare non-follicular-, non-C cell-derived tumors (mucoepidermoid carcinoma (MEC), secretory carcinoma (SC), sclerosing mucoepidermoid carcinoma with eosinophilia (SMECE), cribriform morular thyroid carcinoma (CMTC), thyroblastoma (TB), etc.) [2]. Specific genetic variants have been described in some of these rare, primary tumors of the thyroid (*CRTC1::MAML2* in MEC, *ETV6::NTRK3* in SC, *APC* in CMTC and *DICER1* in TB) [3,4,5,6], but description of these rare tumors is beyond the scope of this article. It is noted that some of these genes are aberrant in more common thyroid tumors and will be discussed further.

As the most frequent thyroid malignancy, papillary thyroid carcinoma (PTC) is most often classic PTC and is predominantly associated with *BRAF* p.V600E [7,8], although most prevalence studies were performed prior to a subset of PTC being relegated to other categories (non-invasive follicular thyroid neoplasm with papillary-like nuclear features (NIFTP), CMTC, well-differentiated tumor of uncertain malignant potential (WDT-UMP)) [8]. Following the publication of The Cancer Genome Atlas characterization of PTC in 2014 [9], which included a prominent subset of formerly classified follicular variant PTC, tumors were largely divided into those that were *BRAF*^V600E^-like and *RAS*-like [10]. A later study by Yoo et al. described the mutational landscape in follicular thyroid adenomas (FTA) and follicular carcinoma (FTC), with reference to PTC, including the previously described *BRAF*-like and *RAS*-like lesions along with a Non-Braf/Non-Ras (NBNR) group of follicular-derived tumors that behaved in an intermediate fashion. This NBNR group included neoplasias encompassing genetic fusion events involving tyrosine kinases (*RET, NTRK, ALK, ROS*, etc.) [11].

In the earlier part of the century, genetic testing was just emerging as a technology to complement standard histology and immunohistochemistry. For thyroid, the concept of routine preoperative molecular testing of thyroid nodules indeterminate for malignancy, following fine needle aspiration biopsy (FNAB), launched in 2010 [12]. Most classic PTC cases (*BRAF*-like) were diagnostic from routine FNAB, yet profiles picked up via indeterminate biopsies, including occasional differentiated thyroid carcinomas (DTC), which include PTC, FTC, and oncocytic thyroid carcinoma (OTC), largely included genetic *RAS*-like variants and those that were NBNR. Further, malignancies, such as MTC, which may be difficult to diagnose via FNAB unless suspected or accompanied by preoperative serum assays, along with rare thyroid malignancies of non-standard cytology, such as cribriform morular thyroid carcinoma, could be assessed [13]. Over the last decade, the preoperative molecular testing of FNAB material has played a significant role in optimizing patient management, triaging for minimal surgery and active surveillance as warranted [1]. However, for advanced thyroid cancers, including anaplastic thyroid carcinoma (ATC), some poorly differentiated thyroid carcinomas (PDTC), MTC and clinically aggressive DTC, naïve/untreated genetic analysis may be essential for determining the best treatment plan, including a subset of patients who may benefit from neoadjuvant therapy [14]. In this review, we focus on the molecular alterations and tests offered in the follicular-derived and C cell-derived thyroid tumors, and we will discuss the potential targeted therapies.

## 2. History of Molecular Tests in Thyroid Diagnostics

Lindsay et al. described PTC nuclear features in thyroid carcinomas in the 1960s, and the subsequent description of the follicular variant of PTC (FVPTC) in the late 1970s by Chen and Rosai were agnostic to molecular alterations. In the following several decades, there was a dramatic shift in the diagnosis of PTC based not only on the presence of architectural papillae in the presence of nuclear features, but also based on the nuclear features alone, even in the absence of architectural features [15]. The histories of molecular alterations and genetic testing in thyroid carcinomas are described in Table 1. In 1987, Fusco et al. first noted *TRK* and *RET* rearrangements in five PTC and their respective metastatic lymph nodes [16]. Inspired by their work, Santoro et al. conducted a study on 286 thyroid tumors (177 PTC, 37 FTC, 15 ATC, 18 MTC, 34 benign thyroid nodules and 5 cases of squamous cell carcinoma and sarcomatoid carcinoma of the thyroid), and found *RET* aberrations exclusively in 19% of PTC (33/177) [17]. In 1989, Lemoine et al. identified *NRAS* p.Q61R and *HRAS* p.Q61R in all stages of the thyroid neoplastic process (FTA, FTC and ATC) [18]. Namba et al. found further genetic variants within *RAS* subtypes (*NRAS, HRAS* and *KRAS*) in codons 12 and 13 in benign and neoplastic thyroid nodules [19]. In 2000, Kroll et al. reported *PAX8::PPARG* fusion in five of eight FTC but not in 20 FTA, 10 PTC or 10 multinodular hyperplasias [20]. Nikiforova et al. found the same fusion event in 8 of 15 (53%) FTC and 2 of 25 (8%) FTA, but not in 35 PTC (including 12 follicular variants), 12 OTC, 12 oncocytic thyroid adenoma (OTA), 2 ATC, 1 PDTC or 16 hyperplastic nodules [21].

At the turn of the twenty-first century, two distinct pathways were emerging in thyroid tumorigenesis, one being *RAS*-associated or *PAX8::PPARG*-related with a predominant follicular growth patten, and another not fully explained by *RET* or *TRK* aberrations. The inverse association between *BRAF* and *RAS* in alternative activation of *MEK/MAPK* signaling in many tumors [35], including colon [36] and ovarian tumors [37], inspired Cohen et al. to explore the role of *BRAF* in the predominantly *RAS*-negative PTC. Their work identified *BRAF* p.T1796A in 69% of PTC (24/35) [22]. Of brief note is that this original nomenclature referred to the nucleotide versus the codon, and the erroneous *BRAF* T1796A (then, *BRAF* p.V599E) eventually came to be recognized as *BRAF* T1799A (now, *BRAF* p.V600E) after the error was realized [38]. Since then, molecular studies have progressed to become a standard and integral part of our understanding of thyroid tumorigenesis and biological potential.

## 3. Fine Needle Aspiration and the Emergence of Molecular Testing Platforms

FNAB with cytological evaluation is the most reliable and commonly used tool for cancer diagnosis in thyroid nodules. The arrival of preoperative ancillary molecular diagnostics for nodules of indeterminate cytology was field-changing for just over the past decade [12]. Prior to molecular platforms for FNAB material, individual gene testing was more common in surgical resections, and testing was more expensive, assayed far fewer genes and worked with less reliable, longer genetic primers [39]. In 2009, Nikiforov et al. conducted a study on FNAB samples from thyroid nodules with Bethesda categories of benign, indeterminate and malignant. Their group analyzed *BRAF* variants (p.V600E, p.K601E), *NRAS*, *KRAS* and *HRAS* along with genetic fusion products *RET::PTC1 (CCDC6::RET), RET::PTC3 (NCOA4::RET)* and *PAX8::PPARG*—genetic changes commonly associated with thyroid neoplasia. Biopsies positive for genetic aberrations were nearly uniformly found to be malignant following excision [23], the results of which have been clarified over the years with broader panels, larger cohorts and refined diagnostic criteria for malignancy [28,31]. The evolution of the most common molecular tests in preoperative FNAB of thyroid nodules is illustrated in Figure 1. This discovery has greatly improved preoperative diagnosis for indeterminate thyroid nodules (ITN), confirming neoplasia and with increasing granularity around risk of malignancy.

In 2012, Alexander et al. conducted a prospective, multicenter validation study of 577 ITN, 413 of which had corresponding histopathological specimens. Using machine learning to interpret mRNA expression of 167 genes through microarray platforms, the Afirma Gene Expression Classifier (GEC) studied ITN and correctly identified 78 of the 85 excised malignant nodules as suspicious with a sensitivity of 92% and a specificity of 52%. The study aimed to eliminate unnecessary thyroid surgeries by ruling out thyroid neoplasms and recommended a conservative approach for most patients with ITN and benign nodules according to GEC results. Afirma GEC was the first rule-out test for thyroid neoplasms [12]. In 2018, the Afirma Genomic Sequencing Classifier (GSC) was developed to better describe the RNA transcriptome with additional sequencing of nuclear and mitochondrial genes and genomic copy number changes, including loss of heterozygosity. This resulted in a more robust genomic test. For ITN with non-oncocytic histology, the sensitivity was 92% and the specificity was 70% [29]. Although Afirma GEC was highly sensitive, certain thyroid tumors, like oncocytic neoplasms, were intentionally called suspicious, resulting in lower specificity for such neoplasms. With the improved GSC, a more comprehensive and updated GEC, Hao et al. used NGS of whole transcriptome RNA sequencing in FNA nodules with oncocytic features to identify mitochondrial and nuclear expression of genes related to oncocytic (formerly Hürthle cell) neoplasms. The specificity of this algorithmic classification increased from 12% to 59% [30].

In 2013, the first version of targeted NGS panel ThyroSeq was published [24]. ThyroSeq used NGS platforms for 12 genes to include 284 hot spot mutations to cover more than 95% of the reported variants in genes associated with thyroid neoplasms. In addition to some mutations detected by GEC (*BRAF*, *RET*, *NRAS*, *KRAS* and *HRAS*), ThyroSeq studied *PIK3CA*, *TP53*, *TSHR*, *PTEN*, *GNAS*, *CTNNB1* and *AKT1*. *PAX8::PPARG* fusion and *RET*- and *TRK*-rearranged genes were not included in ThyroSeq v1 [24]. This panel was expanded as ThyroSeq v2.1 and was published in 2015 to include additional point mutations in *EIF1AX* and *BRAFV601K* and detection of over 40 other gene fusions including *THADA, ALK*, *PAX8::PPARG, TRK1* and *TRK3* genes. Finding *BRAF* mutations or *PPARG, NTRK1, NTRK3* and *ALK* fusions were strong predictors of a higher risk of cancer (approaching 100%) while other mutations like *RAS, PTEN* and *EIF1AX* or *THADA* fusions were associated with a significant but lower risk of cancer [25]. The latest platform of ThyroSeq v3 is the current commercially available test [28].

Other commercially available molecular platforms for preoperative diagnosis of ITN include either a microRNA (miRNA) classifier such as RosettaGX Reveal and mirTHYtype or miRNA and somatic gene mutational platform such as Asuragen [26,27,41].

## 4. Molecular Tests and Clinical Practice

The workflow of thyroid nodules preoperatively is demonstrated in Figure 2. Thyroid FNAB predominantly occur in outpatient clinics and are performed by a range of physicians (endocrinologists, radiologists, surgeons and pathologists). When rapid on-site evaluation (ROSE) is available, adequacy and diagnosis can be made within a few minutes [42], and additional needle pass(es) for ITN can be obtained and sent out for molecular testing to Afirma, ThyroSeq or another desired platform. However, in settings where ROSE is unavailable, specimens are collected for subsequent cytologic and molecular evaluation. In the latter scenario, practices vary among institutions, but a “collect on all” protocol for possible molecular testing seems to be the most efficient for both patients and the lab [43] rather than a return visit and re-biopsy for an indeterminate biopsy. Both Afirma and ThyroSeq provide a send out service as a reflex to ITN FNA. Fresh FNA samples are preferred for optimal and successful testing, but ThyroSeq also offers molecular testing for tissue scrapings from FNA slides and from paraffin-embedded tissue [44].

## 5. Molecular Tests in ITN

Molecular tests were initially performed on ITN as a rule-out malignancy test in order to risk stratify for surgery (negative predictive value) [10]. The indications for these tests have evolved over the past decade. Three highly utilized, commercially available/validated platforms for thyroid nodules in the United States are Afirma Genomic Sequencing Classifier (Afirma GSC)/Afirma Xpression Atlas (XA), ThyGeNEXT/ThyraMIR (MPTX) and ThyroSeq v3.

Afirma GSC uses whole transcriptome RNA analysis using NGS combined with machine learning. Nodules evaluated by Afirma result in a benign or suspicious classification. Initially, samples sent to Afirma are tested for sufficient RNA. Samples are then tested against classifiers to detect parathyroid tissue, MTC, *BRAF* p.V600E, *CCDC6::RET* or *NCOA4::RET*. When all classifiers are negative, GSC analyzes > 10,000 genes and classifies samples as GSC-B (benign) or GSC-S (suspicious). Compared with Afirma GEC, Afirma GSC has enhanced diagnostic accuracy for oncocytic thyroid neoplasms [30]. Introduced in 2019 and updated in 2020, Afirma Xpression Atlas (XA) enumerates mutations in 593 genes, informing 905 variants and 235 fusions in suspicious Afirma GSC, and is extended to include Bethesda V and VI nodules. Comparing Afirma XA testing with whole-transcriptome RNA-, targeted RNA- and targeted DNA-sequencing, Angel et al. reported mutations and gene fusions associated with thyroid nodules with high reproducibility and accuracy. *BRAF* p.V600E was predominantly present in Bethesda V/VI FNA, and *NRAS* and *HRAS* variants were primarily seen in Bethesda III/IV FNA. In addition to point mutations in *BRAF* and *RAS* genes, Afirma XA reported mutations in *TSHR* p.M453T and *SPOP* p.P94R. The two mutations were mostly associated with benign nodules [45,46]. Afirma XA detected many fusions seen in thyroid neoplasms like *PAX8::PPARG*, *ETV6::NTRK3*, *CCDC6::RET*, *NCOA4::RET*, *STRN::ALK*, *AGK::BRAF*, *SND1::BRAF* and *RBPMS::NTRK3*. Fusion of *PAX8::GLIS3* was also detected with Afirma XA, which is associated with hyalinizing trabecular tumor [47,48].

Afirma XA provides information about inherited syndromes with increased likelihood of thyroid malignancy, such as *RET* (multiple endocrine neoplasia 2; MEN2), *PTEN* (*PTEN* hamartoma tumor syndrome; Cowden syndrome), *APC* (*APC*-associated polyposis; familial adenomatous polyposis) and *DICER1* (*DICER1* syndrome). Other inherited mutations in oncogenes or tumor suppressor genes like *BRCA1*, *BRCA2*, *RB1*, *WT1*, *NF2*, *COL3A1*, *TGFBR2*, *TP53* and *MSH2* are also reported with Afirma XA. Since Afirma is not intended to test for germline alterations and mutations may be somatic, patients with clinical suspicion for inherited syndromes should be referred for genetic counseling [40].

MPTX includes the combination of a mutation panel (ThyGeNEXT) and an miRNA risk classifier (ThyraMIR). Results of MPTX are categorized as positive, moderate risk or negative for malignancy. The ThyGeNEXT portion uses targeted NGS for selected DNA gene mutations (*ALK*, *BRAF*, *GNAS*, *HRAS*, *KRAS*, *NRAS*, *PIK3CA*, *PTEN*, *RET* and *TERT* promoter genes) as well as mRNA fusion genes (*ALK*, *BRAF*, *NTRK*, *PPARG*, *RET*, *PAX8*, *TBP*, *USP33* and *THADA*). When there is a strong driver mutation, the results of the test are flagged as positive. Further analysis with the miRNA classifier is carried out when a weak or absent driver mutation is detected. Several growth-suppressing miRNA (miR-29, -138, -139, -155, -204) or growth promoting miRNA (miR-31, -146, -222, -375, -551) are tested in the ThyraMIR portion of MPTX [32]. ThyGeNEXT has been updated to include coexisting driver mutations and increases detection of strong driver mutations (*TERT* promoters or *BRAF* p.V600E) or weak driver mutations (*RAS*), which are both correlated with positive miRNA results [33]. More recently, miR-21 has been added to the MPTX panel (MPTXv2) and shows a decrease in the moderate-risk cohort from 28% to 13% (*p* < 0.001) and an improved diagnostic accuracy of ITN risk stratification [34]. MTC diagnosis showed 100% accuracy with miRNA in the MPTX platform (4 MTC and 26 non-MTC samples) [49].

Thyroseqv3 is a DNA- and RNA-based targeted NGS platform that detects point mutations, gene fusions, copy number alterations and abnormal gene expression in 112 thyroid cancer-related genes. ThyroSeq v3 has a genomic classifier score that is the sum of all genetic alterations detected in the sample, to distinguish benign from malignant nodules and correctly classify most papillary, follicular and oncocytic thyroid neoplasias, MTC and parathyroid lesions with 94% sensitivity, 89% specificity and 92% accuracy [28]. Copy number alterations (CNA) have been reported in approximately 7% of PTC lacking other driver mutations and in other tumors such as oncocytic thyroid neoplasms, which show a nearly homozygous genome [50,51,52]. ThyroSeq v3 has a sensitivity of 93% and a specificity of 69% for oncocytic thyroid nodules. ThyroSeq v3 classifies nodules based on the likelihood of malignancy into six scores with a proposed cutoff of 1.5 for malignant lesions [28]. However, the ultimate arbiter is surgical pathology diagnosis and patient outcomes, in the context of molecular changes, combined with any incremental pathophysiologic data that may enhance this new baseline.

MPTX, Afirma GSC and ThyroSeq v3 were primarily designed as negative predictive value tests, although the use of molecular analysis in thyroid nodules has evolved. In one meta-analysis, ThyroSeq v3 and Afirma GSC showed optimal results to exclude malignancy. Both ThyroSeq v3 and Afirma GSC were not superior to ThyroSeq v2 in ruling out malignancy. MPTX reported similar results [53]. In two other meta-analyses by Lee et al. and Livhits et al., diagnostic performances of Afirma GSC and ThyroSeq v3 were not statistically different, and both showed similar specificity and reduced unnecessary diagnostic surgeries in 49% of ITN [54,55].

Regardless of the molecular platform used, the goal is to provide sufficient utility for optimal patient management. Lacking concomitant clinical concerns, a benign molecular finding in an ITN may be clinically observed. Although advances have been made in the molecular testing for oncocytic thyroid nodules and *RAS*-like genetic variant tumors (FTA, NIFTP or FTC), histology is still the gold standard in examining the entire periphery of the nodule to assess for capsular or angioinvasion, and such nodules are preferably treated with surgery, typically hemithyroidectomy/lobectomy barring concomitant clinical circumstances. For patients with higher-risk genetic or clinical profiles (radiologically high-risk +/− lateral neck disease), more comprehensive upfront surgery may be considered. Molecular testing is widespread in the United States; however, in most of the world, cytologic diagnosis alone, if available, is the primary determinant of risk of malignancy in thyroid nodules. There have been cost-effectiveness studies conducted that show great value in the cost of molecular testing versus unnecessary thyroid surgery for molecularly determined benign nodules [56]. In particular, clinical management of thyroid nodules in under-resourced populations is not well-studied internationally, with limited data for underserved areas of the United States [57]. As an alternative, immunocytochemistry on fine needle aspiration preparations or immunohistochemistry on core biopsies can provide some limited insight at a lower cost in some specific situations [58]. However, without molecular testing, the decision for surgery, including extent of surgery, is confined to the clinical scenario with possible input from FNAB interpretation, and this may result in a greater incidence of total thyroidectomy for a single ITN in up to 90% of cases depending upon geography [59].

## 6. Cancer Prognosis in the Era of Molecular Testing

Molecular alterations in thyroid neoplasms play an essential role in classification, prognosis and response to radioactive iodine (RAI). In 2014, The Cancer Genome Atlas (TCGA) consortium extensively investigated the molecular landscape of its selected PTC cohort and the distribution of data largely fell into a binary set of molecular alterations favoring a *BRAF*^V600E^-like or *RAS*-like phenotype. Although this cohort reduced the “black matter” in PTC from 25% to 3.5%, fusion genes like *RET*, *NTRK1/3* and *ALK* showed neutral (non-*BRAF-*/non-*RAS*-like; NBNR) to weak *BRAF*-like signaling, thus questioning the correlation between such mutations and *BRAF* genes [9]. Focusing on FTA and minimally invasive FTC, Yoo et al. classified the molecular subtypes into three groups, *BRAF*-like, *RAS*-like and NBNR-mutated tumors. The *BRAF*-like mutated tumors were associated with a higher stage at presentation, larger tumors and lateral neck metastasis, compared with the *RAS*-like and NBNR groups. NBNR included tumors with *DICER1, EIF11AX, PTEN* and *IDH1 mutations* and *PAX8::PPARG* fusion [11].

In thyroid nodules preoperatively classified as malignant (Bethesda VI), *BRAF* p.V600E mutation was identified in 65–75% of tumors, followed by *TERT* promoter mutations that were seen in 11% [60]. The presence of both *BRAF* p.V600E and *TERT* promoter was significantly correlated with patient age, extrathyroidal extension, lymph node metastasis, distant metastasis, disease stage, tumor recurrence and mortality [61]. The presence of both mutations in PTC robustly predicts loss of radioiodine avidity in these tumors [62].

In one study, suspicious nodules (Bethesda V) were analyzed using ThyroSeq v3, and were categorized into low-, intermediate- and high-risk. Low-risk tumors were mostly associated with *RAS*-like mutations and focal chromosomal-type CNA, and intermediate risk comprised the *BRAF*-like mutations and genome haploidization-type CNA. The presence of *TP53* and *TERT* promoter mutations was considered a high-risk mutation. High-risk mutations were associated with ATC, PDTC and tall cell PTC. Follow-up data on the high-risk cases showed more frequent recurrences than low-risk cases with limited risk of recurrence, affirming the predictive value of molecular profiling [63]. When PDTC was studied using ThyroSeq v3, high-, intermediate- and low-risk tumors were 60%, 23% and 17%, respectively, and high-risk tumors were associated with distant metastasis and worse overall survival, compared with low-risk, where no patient died of disease [64]. Genetic studies in higher grade tumors may be helpful in finding molecular targets as neoadjuvant therapy.

## 7. Molecular Identification of Targetable Alterations in Thyroid Cancer

The most common medications used for targetable mutations in thyroid carcinoma are summarized in Table 2. In the setting of advanced thyroid carcinomas, molecular testing is essential for identifying potential therapeutic options, including those tumors that were previously tested but have acquired a behavioral change. High-grade thyroid carcinomas, including PDTC, MTC and ATC, often present with advanced disease; however, 15% of DTC may also be locally aggressive, necessitating additional treatment, including RAI. In a study of metastatic non-anaplastic thyroid carcinomas (PTC, FTC, PDTC), Durante et al. found that only 30% of patients showed RAI uptake with remission, 39% demonstrated RAI uptake with no remission and in the remaining 31%, no RAI was detected. RAI treatment plays a crucial role in the treatment of advanced thyroid carcinomas. Overall survival rate at 10 years post-RAI was 92% in patients who achieved a negative study compared with 19% in patients who did not [65]. RAI-resistant patients can be treated with cytotoxic chemotherapy as a monotherapy or a combination of more than one agent; however, the response is partial, with troublesome side effects with a relatively narrow therapeutic index [66]. In this scenario, with the most advanced tumors resistant to traditional treatments like RAI or presenting with distant metastasis, expansion of treatment options to include therapies targeting specific genetic variants is promising.

Most thyroid neoplasia is related to activation of the mitogen-activated protein kinase (MAPK) and the PI3K pathways. When no targetable mutation is identified (*NTRK*, *ALK*, *RET* or *BRAF*) or the tumor is not sequenced, the antiangiogenic multi-targeted kinase inhibitors (aaMKIs) Sorafenib, Lenvatinib and Cabozantinib may be used. AaMKIs have been approved by the US Food and Drug Administration (FDA) for the treatment of aggressive RAI-resistant PTC or FTC. Both Sorafenib and Lenvatinib can be used as first-line treatment in RAI-resistant thyroid carcinomas. Lenvatinib is superior to Sorafenib in achieving longer progression-free median survival. Previously approved by the FDA for the treatment of MTC, Cabozantinib has been authorized as a second-line treatment when tumors fail to respond to Lenvatinib or Sorafenib, and it showed improved progression-free survival. Although aaMKIs slow the progression of disease, their toxicity profiles limit their clinical use due to nonspecific targeting [68,69,70,71,81].

A mutation-specific inhibitor should be considered when a specific driver mutation is identified (*RET*, *NTRK*, *ALK* or *BRAF*). Selpercatinib and Pralsetinib inhibit *RET* in MTC, but they can also block the *RET* fusion protein-mediated signaling found in PTC [80,82]. Although therapeutic resistance has been reported [84,85], *RET* inhibitors are better tolerated and more specific than aaMKIs.

In *TRK* fusion-positive thyroid cancers, Larotrectinib demonstrates rapid and durable disease control and a favorable safety profile in patients with advanced disease [83]. *ROS1* inhibitors such as Entrectinib have also been used in one patient with *ROS1* mutation thyroid carcinomas with liver metastasis and showed efficacy. Although this mutation is rare and no randomized clinical trials have been conducted on thyroid due to the rarity of this fusion, Entrectinib might be used in advanced cases with intractable disease [76].

Earlier reports of Dabrafenib and Vemurafenib show partial response to treatment in *BRAF* p.V600E-mutated PTC [72,73]. More recently, enhanced response to RAI and increased iodine uptake has been achieved with pretreatment of RAI-resistant PTC with Dabrafenib and Vemurafenib. This approach may represent a new potential treatment option for patients with RAI-resistant tumors [74,75]. Although *ALK* fusion represents <1% of thyroid carcinoma, it is more frequently seen in PDTC. There is no current FDA-approved treatment for *ALK* kinase fusion-related thyroid carcinomas; however, ALK inhibitors may be used for clinical trials or off-label treatment in advanced cases [67,86].

Although most thyroid carcinomas are microsatellite-stable and show low tumor mutational burden (TMB), a subset of follicular carcinomas (~2.5%) may show microsatellite instability or high TMB [87]. Programmed death (PD-1) inhibitors (including Pembrolizumab) can be also used, based on the agnostic approval for solid tumors [79].

ATC is a rare but lethal subtype of thyroid carcinoma with a historical 3–4 month median survival rate [88,89], a survival rate significantly improved with targeted therapeutics, now greater than one year [90]. In 58% of ATC, there is a history or a concurrent DTC, and in the remaining 42%, ATC is considered de novo. ATC tend to accumulate multiple mutations in genes readily present in DTC like *BRAF* or *RAS*, or kinase fusion-related thyroid carcinomas (*NTRK*, *RET* or *ALK*), in addition to higher risk genes like *TERT* promoter, *TP53*, *PIK3CA* and *PTEN* [91,92]. The treatment of ATC has been advanced in the past decade to include the FDA-approved usage of *BRAF/MEK* inhibitors (Dabrafenib/Trametinib) in *BRAF* p.V600E-mutated ATC [77,78]. The first report on vemurafenib to treat *BRAF* p.V600E-associated ATC was in 2013 [93]. The detection of *BRAF* p.V600E is crucial and time-sensitive in determining the course of treatment in ATC. Old and traditional tests like NGS (2–3 weeks) may delay treatment and rapid tests are needed to assess *BRAFV* p.V600E mutation and start treatment with *BRAF/MEK* inhibitors. *BRAFV* p.600E hot spot mutation can be detected either by mutation-specific immunohistochemistry on paraffin-embedded blocks of needle core biopsies or a cell block of FNA [94]. *BRAF* IHC is quite sensitive but may be accompanied by background staining. Results of *BRAF* IHC turn around is fast; it is <24 h for in-house testing [94]. *BRAF* p.V600E DNA can be detected using rapid PCR assay of an extract of paraffin blocks (2–3 days) or from the peripheral blood NGS (cell free DNA) [95,96]. Pyrosequencing is also an effective method for detecting the *BRAF* p.V600E mutation in FNA samples [97], and innovative methods for detection to obtain the *BRAF* p.V600E in less than 1 day have been developed [98].

Regardless of the method used to evaluate for *BRAF* p.V600E mutation, *BRAF*-directed therapy should be started as soon as possible to induce rapid and substantial disease regression. Advanced tumor stage ATC, which has been considered inoperable, can now profit from *BRAF* inhibitors alone or *BRAF/MEK* inhibitors. Surgery is a novel option in such patients with significant tumor reduction up to 100% locoregional control, 83% and 32% survival rate at one and two years, respectively [77,78,99]. Despite the actionable targeted therapies for *BRAF* p.V600E, *MEK*, *NTRK*, *ALK*, *RET* and *ROS1*, acquired resistance may develop following an initial response and biopsy of resistant lesions, and molecular testing may be appropriate in assisting with new therapeutic strategies [100].

MTC is a non-follicular origin thyroid tumor that may occur sporadically or be inherited, as part of the Multiple Endocrine Neoplasia 2 (MEN 2) syndrome and Familial Medullary Thyroid Carcinoma (FMTC) syndrome. *RET* mutations have been described in both germline and sporadic cases, while *RAS* mutations have been found only in sporadic cases [101]. At presentation, 58% of MTC presented with lymph node metastasis and 6% with distant metastasis [102]. Inherited MTC are associated with an autosomal dominant mutation in the *RET* protooncogene (40% of MTC). All patients diagnosed with MTC are candidates for genetic counselling with family testing for potential inherited *RET* mutations. In 47% of sporadic patients, *RET* mutations are sporadic, and no further treatment except follow-up is needed. *RAS* accounts for 26% of sporadic MTC (*HRAS, KRAS* and, less frequently, *NRAS*). In 19% of sporadic tumors, neither *RET* nor *RAS* mutations were identified [103]. One study showed that somatic *RET*-mutated MTC (exon 15 and 16) correlated with multiple lymph node metastasis, multiple tumors and presented with a higher clinical stage and increased metastatic rate compared with non-exon 15/16 and non-*RET* mutated MTC [104]. Cabozantinib and Vandetanib are *RET*-targeted therapies that have been approved by the FDA for progressive locally advanced and/or metastatic *RET*-mutated MTC and show slow progression of disease. Both medications are associated with significant toxicity due to a narrow therapeutic window and non-selective kinase inhibition [105,106]. Although non-MTC *RET*-mutated tumors have been associated with resistance to Vandetanib and Cabozantinib, *RET*-mutated MTC at codon V804 (gatekeeper mutation) showed indolent disease without classic MEN 2A features [107]. With more selective *RET* inhibitors like Selpercatinib, a considerable response was recorded in patients initially treated with Vandetanib and/or Cabozantinib, and it was recently approved by the FDA for treating advanced *RET*-mutated MTC. The most recent selective *RET* inhibitor approved by FDA is Pralsetinib. Advanced or metastatic MTC patients treated with Pralsetinib show improved response rates (65%), with *RET*-mutated tumors, including MKI-resistant and codon V804 mutations. Fewer side effects are recorded and mostly are reversable, which makes this medication a potential therapeutic for inoperable or advanced MTC [82].

In summary, molecular testing has become a standard tool in thyroid nodule management. With its reduced costs, there are increasing applications for its use, particularly in the space of emerging small, targetable therapeutics for advanced thyroid cancers, particularly those with variants in *BRAF*, *RET* or kinase fusions. Molecular testing frequently falls short, as does surgical pathology, in discerning follicular-patterned thyroid adenomas from carcinomas, and this remains an opportunity for refined risk stratification and therapeutic planning following molecular testing in these lesions.

## Figures and Tables

**Figure 1 genes-14-01314-f001:**
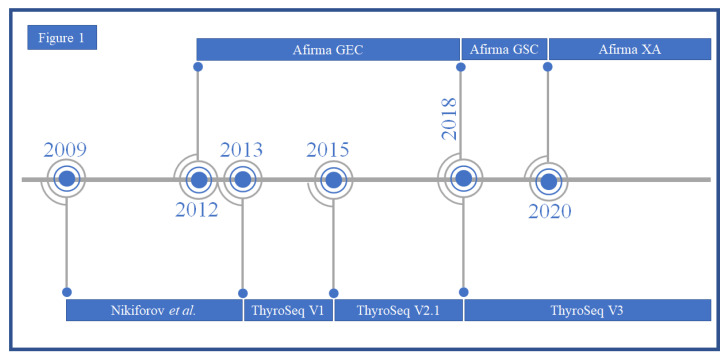
Timeline of the two most common molecular tests for preoperative evaluation of thyroid nodules for fine needle aspiration biopsy; Afirma and ThyroSeq. Nikiforov et al. published the earliest molecular testing on FNA in 2009 [23] but the test became available in 2013 as ThyroSeq V1 [24]. ThyroSeq V2.1 followed in 2015 [25], and the current commercially available test is ThyroSeq V3 [20]. Afirma GEC started in 2012 [12] and updated as Afirma GSC in 2018 [29]. The latest version that is currently available is Afirma XA [40]. GEC: Gene Expression Classifier, GSC: Genetic Sequencing Classifier and XA: Xpression Atlas.

**Figure 2 genes-14-01314-f002:**
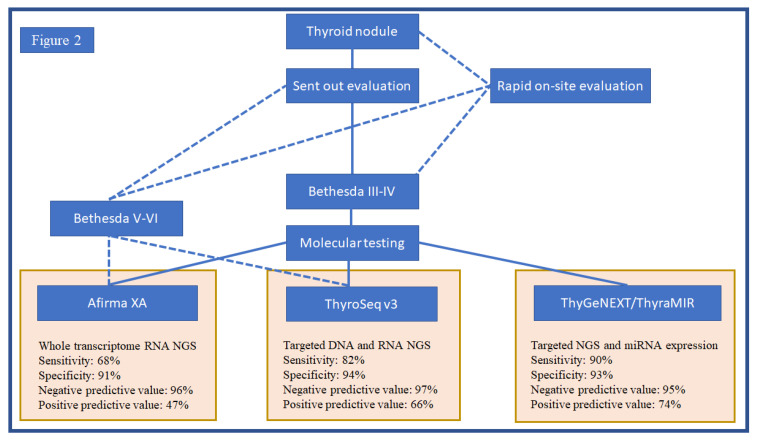
Algorithm for preoperative evaluation of thyroid nodules following fine needle aspiration biopsy (FNAB). Nodules of Bethesda III and IV (indeterminate thyroid nodules) are sent for molecular testing for further classification. Nodules of Bethesda V and VI (suspicious and malignant) can be sent for Afirma XA or ThyroSeq v3 to study the tumor landscape and potential high-risk mutations (TERT promotor, TP53, etc.). Most FNAB are sent out and most labs have a reflex for molecular testing for Bethesda III or IV. When rapid on-site evaluation is available, samples can be sent directly for molecular testing. NGS: Next Generation Sequencing, XA: Xpression Atlas, RNA: ribonucleic acid, DNA: Deoxyribonucleic acid and miRNA: micro ribonucleic acid.

**Table 1 genes-14-01314-t001:** History of Molecular Diagnostics.

Year Authors	Molecular Alterations or Genetic Testing	Diagnosis
1987 Fusco et al. [16]	*TRK* and *RET* rearrangements	PTC
1989 Lemoine et al. [18]	*NRAS* p.Q61R and *HRAS* p.Q61R	Thyroid neoplastic process (FTA, FTC and ATC)
1990 Namba et al. [19]	*NRAS, HRAS* and *KRAS* at codons 12 and 13	Benign and neoplastic thyroid nodules
1992 Santoro et al. [17]	*RET* aberrations	PTC
2000 Kroll et al. [20]	*PAX8::PPARG* fusion	FTC
2002 Nikiforova et al. [21]	*PAX8::PPARG* fusion	FTC and FTA
2003 Cohen et al. [22]	*BRAF* p.T1796A point mutation	PTC
2009 Nikiforov et al. [23]	*BRAF* (p.V600E, p.K601E), *NRAS, KRAS* and *HRAS* and *RET::PTC1, RET::PTC3* and *PAX8::PPARG* gene fusion	Bethesda III and IV
2012 Alexander et al. [12]	Afirma GEC: (using machine learning to interpret the expression of mRNA of 167 genes through microarray platforms)	Bethesda III and IV
2013 Nikiforova et al. [24]	ThyroSeq V1: NGS (*BRAF, RET, NRAS, KRAS, HRAS, PIK3CA, TP53, TSHR, PTEN, GNAS, CTNNB1* and *AKT1*)	Bethesda III and IV
2014 The Cancer Genome Atlas (TCGA) [9]	*BRAF*^V600E^-like or *RAS*-like phenotype	PTC including follicular variant
2015 Yoo et al. [11]	*BRAF*-like, *RAS*-like and non-*BRAF-*/non-*RAS*-like (NBNR)	FTA and minimally invasive FTC
2015 Nikiforov et al. [25]	ThyroSeq V2.1: ThyroSeq V1 and *EIF1AX* and *BRAFV601K* and 40 gene fusions including *THADA, ALK*, *PAX8::PPARG, TRK1* and *TRK3*	Bethesda III and IV
2016 Wylie et al. [26]	RosettaGX Reveal (miRNA classifier)	Bethesda III and IV
2016 Benjamin et al. [27]	Asuragen (miRNA and somatic gene mutational platform)	Bethesda III and IV
2018 Nikiforova et al. [28]	ThyroSeq V3: NGS of 112 thyroid cancer-related genes	Bethesda III and IV
2018 Patel et al. [29]	Afirma GSC (describing RNA transcriptome with additional sequencing of nuclear and mitochondrial genes, changes in genomic copy number including loss of heterozygosity)	Bethesda III and IV
2019 Hao et al. [30]	Afirma GSC (NGS of whole transcriptome RNA sequencing)	Oncocytic cell neoplasms
2019/2020 Hu et al. [31]	Afirma Xpression Atlas (enumerates mutations in 593 genes informing 905 variants and 235 fusions in suspicious Afirma GSC)	Bethesda III, IV, V and VI
2020 Sistrunk et al. [32] Jackson et al. [33] Finkelstein et al. [34]	MPTX (ThyGeNEXT/ThyraMIR): NGS (*ALK*, *BRAF*, *GNAS*, *HRAS*, *KRAS*, *NRAS*, *PIK3CA*, *PTEN*, *RET* and *TERT* promoter genes) as well as mRNA fusion genes (*ALK*, *BRAF*, *NTRK*, *PPARG*, *RET*, *PAX8*, *TBP*, *USP33* and *THADA*) and miRNA- (21, 29, 31, 138, 139, 155, 146, 204, 222, 375, 551)	Bethesda III and IV

PTC: papillary thyroid carcinoma, FTA: follicular thyroid adenoma, FTC: follicular thyroid carcinoma, ATC: anaplastic thyroid carcinoma, mRNA: messenger ribonucleic acid, miRNA: micro ribonucleic acid, NGS: Next Generation Sequencing, GEC: Gene Expression Classifier and GSC: Genetic Sequencing Classifier.

**Table 2 genes-14-01314-t002:** FDA-approved medications for different subtypes of thyroid carcinoma.

Medication	Molecular Target	Mechanism of Action	Year	Reference
Vandetanib	Inhibits *RET* in MTC	Selective *RET* TKI	2012	Degrauwe et al. [10]
Alectinib *	*ALK1* fusion-positive thyroid carcinoma	*ALK*-TKI	2014	Kelly et al. [67]
Sorafenib	Aggressive RAI-resistant PTC or FTC	Anti-angiogenic multi-targeted kinase inhibitor (aaMKI)	2014 2015	Brose and Nutting et al. [68] Ferrari, Politti et al. [69]
Lenvatinib	Aggressive RAI-resistant PTC or FTC	Anti-angiogenic multi-targeted kinase inhibitor (aaMKI)	2015 2021	Ferrari and Elia et al. [70] Schlumberger et al. [71]
Dabrafenib	*BRAF* p.V600E-mutated PTC	*BRAF* inhibitor	2015 2016	Falchook et al. [72] Brose and Cabanillas et al. [73]
Dabrafenib and Vemurafenib	*BRAF* p.V600E-mutated PTC	*BRAF* inhibitor	2015 2019	Rothenberg et al. [74] Dunn et al. [75]
Entrectinib	*ROS1* fusion-positive thyroid carcinoma	Multikinase inhibitor (*NTRK1/2/3, ROS1* and *ALK*)	2017	Liu et al. [76]
Dabrafenib/ Trametinib	*BRAF* p.V600E-mutated ATC	*BRAF/* *MEK* inhibitors	2018 2022	Subbiah and Kreitman et al. [77] Subbiah and Kreitman et al. [78]
Pembrolizumab	Solid tumors including thyroid tumors	Programmed death (PD-1) inhibitors	2019	Marcus et al. [79]
Selpercatinib	-Inhibits *RET* in MTC -*RET* fusion-positive thyroid carcinoma	Selective *RET* TKI	2020	Wirth et al. [80]
Cabozantinib	Second line for treatment of aggressive RAI-resistant PTC or FTC	Antiangiogenic multi-targeted kinase inhibitor (aaMKIs)	2021	Brose and Robinson et al. [81]
Pralsetinib	-Inhibits *RET* in MTC -*RET* fusion-positive thyroid carcinoma	TKI	2021	Subbiah and Hu et al. [82]
Larotrectinib	*TRK* fusion-positive thyroid carcinoma	Selective *TRK* inhibitor	2022	Waguespack et al. [83]

* Approved for non-small cell lung carcinoma. FDA: the Food and Drug Administration, RAI: Radioactive iodine, PTC: Papillary thyroid carcinoma, FTC: Follicular thyroid carcinoma, ATC: Anaplastic thyroid carcinoma and MTC: Medullary thyroid carcinoma. TKI: tyrosine kinase inhibitor.

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
