# Peer review of "Update on Molecular Diagnostics in Thyroid Pathology: A Review"

_genes, 2023, doi:10.3390/genes14071314_

Round 1

Reviewer 1 Report

The manuscript “Update on Molecular Diagnostics in Thyroid Pathology” is a nice and well written review presenting in detail the available molecular tests in thyroid diagnostics and its contribution to improve diagnosis and management of thyroid lesions. It also presents the evolution of molecular testing starting from 1987 up today. In addition, it highlights the diagnostic role of fine needle aspiration in conjunction with the emerging molecular testing platforms. Finally, the FDA-approved medications for different subtypes of thyroid carcinoma are listed and their molecular pathways are described.

My only concern is the following:

Abstract, Page 1, Lines 12,13

The authors state “…  however, in 30% of cases, results are indeterminate for malignancy, and further testing is needed.”

The original paper by Haugen BR et al., quoted in the introduction (Page 1, Line 34) states: “However, 15 to 30% of aspirations yield indeterminate cytologic findings”.

Also, Alexander et al. N Engl J Med 367:705-715, 2012. doi: 10.1056/NEJMoa1203208, report: “Approximately 15 to 30% of thyroid nodules evaluated by means of fine-needle aspiration are not clearly benign or malignant. Patients with cytologically indeterminate nodules ……”.

Similar results are reported by Marti JL et al. Ann Surg Oncol. 22(12):3996-4001, 2015. doi: 10.1245/s10434-015-4486-3 : “By FNA, 10–30 % of biopsied thyroid nodules are categorized as “indeterminate.”

Suggestion: Please revise this sentence as: “... however, in up to 30% of cases, results are indeterminate for malignancy, and further testing is needed.”

Author Response

Suggestion: Please revise this sentence as: “... however, in up to 30% of cases, results are indeterminate for malignancy, and further testing is needed.”

Response: We thank the reviewer and altered the manuscript to read “in up to 30% of cases”. This indeed is less hyperbolic and in line with the truth as published by the referenced articles.

Reviewer 2 Report

The article is well written. Due to the high praevalence of thyroid nodules and their relatively rare malignancy any improvement in rule-in and rule-out testing is welcome. The authors focus on molecular diagnostics and the commercially tests available.

Nevertheless I would recommend to include a few sentences on the other diagnostic tools to sort out malignancy (e.g. sonography/T-IRADS, elastography, scintigraphy).

As the authors state, the commercially available tests are quite popular in America, but rarely used in the rest of the world. As Cells is a international read publication, could you give some information on the cost efficacy? And on alternative ways to gain knowledge about the dignity of the laesion? The most relevant mutations are BRAF p.V600E, RET fusions, NTRK, ALK, where IHC is an option as well.

I suggest to add a conclusion with a short summary which molecular results rule-out and which rule-in malignancy, what relevance molecular diagnostics have for the problematic entities - FTA vs FTC and microcarcinoma.

Please check for the correct reporting on labeling of canonical BRAF mutations: it writes BRAF T1796A (e.g. table 1) as it is reported by codons (thymine-> adenine exchange at nucleotide 1796). The corresponding acid change positioning is BRAF p.V600E (or with another annotation p.V599E), other than labeled e.g. in lines 389 and 390. Maybe you could insert a short explanation on this term for our readers, as the difference of annotations and positions

The median survival rate of ATC is nowadays longer than the restrospective results of 3-4 months (line 380) as stated in a recent review (Lang et al. Thyroid Research (2023) 16:5, https://doi.org/10.1186/s13044-023-00147-7). With targeted therapies it exceeds 1 year (Lenvatinib+Pembrolizumab and in BRAF p.V600E mutated Dabrafenib+Trametinib or Vemurafenib). I would think you rather refer to this.

The first report on the effect of vemurafenib in BRAF mutated ATC was the following: Rosove MH, Peddi PF, Glaspy JA. BRAF V600E inhibition in anaplastic thyroid cancer. N Engl J Med. 2013;368:684–5.

In line 69: MTC without brackets.

Author Response

ancillary non-pathological evaluation of thyroid nodules, but this referred to super briefly (not the specific methods  -- TIRADS, etc) in the abstract. I was searching for a place to insert this into the document, and it only served to make this already long review more cumbersome. Since the review is entitled update on molecular diagnostics, I am going to continue to leave this out, although I do think the idea is part of the broader topic. I have included the rest of the reviewers great suggestions into the manuscript.

As the authors state, the commercially available tests are quite popular in America, but rarely used in the rest of the world. As Genes is an internationally read publication, could you give some information on the cost efficacy? And on alternative ways to gain knowledge about the dignity of the lesion? The most relevant mutations are BRAF p.V600E, RET fusions, NTRK, ALK, where IHC is an option as well.

A sentence is added (with reference) regarding cost efficacy for thyroid molecular testing in indeterminate nodules.

Hu QL, Schumm MA, Zanocco KA, Yeh MW, Livhits MJ, Wu JX. Cost analysis of reflexive versus selective molecular testing for indeterminate thyroid nodules. Surgery. 2022 Jan;171(1):147-154. doi: 10.1016/j.surg.2021.04.050. Epub 2021 Jul 17. PMID: 34284895.

An additional sentence is added (with reference) regarding use of IHC and immunocytochemistry in FNA and core biopsy specimens where no molecular testing is available.

Xiong Y, Li X, Liang L, Li D, Yan L, Li X, Di J, Li T. Application of biomarkers in the diagnosis of uncertain samples of core needle biopsy of thyroid nodules. Virchows Arch. 2021 Nov;479(5):961-974. doi: 10.1007/s00428-021-03161-y. Epub 2021 Jul 26. PMID: 34308507; PMCID: PMC8572826.

I suggest to add a conclusion with a short summary which molecular results rule-out and which rule-in malignancy, what relevance molecular diagnostics have for the problematic entities - FTA vs FTC and microcarcinoma.

The concluding paragraph was updated to specify the useful genes picked up on molecular testing that of confirmational/treatment benefit from molecular testing and further, stating the shortfalls for FTA vs FTC that molecular diagnostics is not as helpful for.

Please check for the correct reporting on labeling of canonical BRAF mutations: it writes BRAF T1796A (e.g. table 1) as it is reported by codons (thymine-> adenine exchange at nucleotide 1796). The corresponding acid change positioning is BRAF p.V600E (or with another annotation p.V599E), other than labeled e.g. in lines 389 and 390. Maybe you could insert a short explanation on this term for our readers, as the difference of annotations and positions.

Thank you to the reviewer for pointing this out. We really did skip this part of the discussion, as we have been a bit wordy already in this review, but it is confusing, so we have slipped in a short explanation, as recommended, and a new reference.

Rowe LR, Bentz BG, Bentz JS. Detection of BRAF V600E activating mutation in papillary thyroid carcinoma using PCR with allele-specific fluorescent probe melting curve analysis. J Clin Pathol. 2007 Nov;60(11):1211-5. doi: 10.1136/jcp.2006.040105. Epub 2007 Feb 13. PMID: 17298986; PMCID: PMC2095462.

The median survival rate of ATC is nowadays longer than the restrospective results of 3-4 months (line 380) as stated in a recent review (Lang et al. Thyroid Research (2023) 16:5, https://doi.org/10.1186/s13044-023-00147-7). With targeted therapies it exceeds 1 year (Lenvatinib+Pembrolizumab and in BRAF p.V600E mutated Dabrafenib+Trametinib or Vemurafenib). I would think you rather refer to this.

Thank you to the reviewer. We have kept the historical survival and supplemented it with the data from Lang et al. to note improved survival.

Lang M, Longerich T, Anamaterou C. Targeted therapy with vemurafenib in BRAF(V600E)-mutated anaplastic thyroid cancer. Thyroid Res. 2023 Mar 1;16(1):5. doi: 10.1186/s13044-023-00147-7. PMID: 36855200; PMCID: PMC9976495.

The first report on the effect of vemurafenib in BRAF mutated ATC was the following: Rosove MH, Peddi PF, Glaspy JA. BRAF V600E inhibition in anaplastic thyroid cancer. N Engl J Med. 2013;368:684–5.

Thank you. We have added this reference in the ATC area.

Rosove MH, Peddi PF, Glaspy JA. BRAF V600E inhibition in anaplastic thyroid cancer. N Engl J Med. 2013 Feb 14;368(7):684-5. doi: 10.1056/NEJMc1215697. PMID: 23406047.

In line 69: MTC without brackets.

Thank you. I don’t know how that happened, but the brackets are now gone.

Round 2

Reviewer 1 Report

The authors have made the indicated changes.

Reviewer 2 Report

Please remove the "p." for the thymine (T) to adenine (A) transversion
mutation (BRAF T1799A or formerly T1796A instead of BRAF p.T1799A / p.T1796A) as the p. indicates the proteine substitution of valine with glutamate in codon 600 (p.V600E, formerly p.V599E)

Otherwise nice work! I enjoyed reviewing.